# Time-series Generative Adversarial Networks

**Jinsung Yoon**[*]
University of California, Los Angeles, USA
jsyoon0823@g.ucla.edu

**Daniel Jarrett**[*]
University of Cambridge, UK
daniel.jarrett@maths.cam.ac.uk

**Mihaela van der Schaar**
University of Cambridge, UK
University of California, Los Angeles, USA
Alan Turing Institute, UK
mv472@cam.ac.uk, mihaela@ee.ucla.edu

## Abstract

A good generative model for time-series data should preserve *temporal dynamics*, in the sense that new sequences respect the original relationships between variables across time. Existing methods that bring generative adversarial networks (GANs) into the sequential setting do not adequately attend to the temporal correlations unique to time-series data. At the same time, supervised models for sequence prediction—which allow finer control over network dynamics—are inherently deterministic. We propose a novel framework for generating realistic time-series data that combines the flexibility of the unsupervised paradigm with the control afforded by supervised training. Through a learned embedding space jointly optimized with both supervised and adversarial objectives, we encourage the network to adhere to the dynamics of the training data during sampling. Empirically, we evaluate the ability of our method to generate realistic samples using a variety of real and synthetic time-series datasets. Qualitatively and quantitatively, we find that the proposed framework consistently and significantly outperforms state-of-the-art benchmarks with respect to measures of similarity and predictive ability.

## 1 Introduction

What is a good generative model for time-series data? The temporal setting poses a unique challenge to generative modeling. A model is not only tasked with capturing the distributions of features *within* each time point, it should also capture the potentially complex dynamics of those variables *across* time. Specifically, in modeling multivariate sequential data $\mathbf{x}_{1:T} = (\mathbf{x}_1, ..., \mathbf{x}_T)$, we wish to accurately capture the conditional distribution $p(\mathbf{x}_t|\mathbf{x}_{1:t-1})$ of temporal transitions as well.

On the one hand, a great deal of work has focused on improving the temporal dynamics of autoregressive models for sequence prediction. These primarily tackle the problem of compounding errors during multi-step sampling, introducing various training-time modifications to more accurately reflect testing-time conditions [1, 2, 3]. Autoregressive models explicitly factor the distribution of sequences into a product of conditionals $\prod_t p(\mathbf{x}_t|\mathbf{x}_{1:t-1})$. However, while useful in the context of forecasting, this approach is fundamentally deterministic, and is not truly *generative* in the sense that new sequences can be randomly sampled from them without external conditioning. On the other hand, a separate line of work has focused on directly applying the generative adversarial network (GAN) framework to sequential data, primarily by instantiating recurrent networks for the roles of generator and discriminator [4, 5, 6]. While straightforward, the adversarial objective seeks to model $p(\mathbf{x}_{1:T})$ directly, without leveraging the autoregressive prior. Importantly, simply summing

---

[*] indicates equal contribution

the standard GAN loss over sequences of vectors may not be sufficient to ensure that the dynamics of the network efficiently captures stepwise dependencies present in the training data.

In this paper, we propose a novel mechanism to tie together both threads of research, giving rise to a generative model explicitly trained to preserve temporal dynamics. We present Time-series Generative Adversarial Networks (TimeGAN), a natural framework for generating realistic time-series data in various domains. First, in addition to the *unsupervised* adversarial loss on both real and synthetic sequences, we introduce a stepwise *supervised* loss using the original data as supervision, thereby explicitly encouraging the model to capture the stepwise conditional distributions in the data. This takes advantage of the fact that there is more information in the training data than simply whether each datum is real or synthetic; we can expressly learn from the transition dynamics from real sequences. Second, we introduce an *embedding network* to provide a reversible mapping between features and latent representations, thereby reducing the high-dimensionality of the adversarial learning space. This capitalizes on the fact the temporal dynamics of even complex systems are often driven by fewer and lower-dimensional factors of variation. Importantly, the supervised loss is minimized by jointly training both the embedding and generator networks, such that the latent space not only serves to promote parameter efficiency—it is specifically conditioned to facilitate the generator in learning temporal relationships. Finally, we generalize our framework to handle the mixed-data setting, where both static and time-series data can be generated at the same time.

Our approach is the first to combine the flexibility of the unsupervised GAN framework with the control afforded by supervised training in autoregressive models. We demonstrate the advantages in a series of experiments on multiple real-world and synthetic datasets. Qualitatively, we conduct t-SNE [7] and PCA [8] analyses to visualize how well the generated distributions resemble the original distributions. Quantitatively, we examine how well a post-hoc classifier can distinguish between real and generated sequences. Furthermore, by applying the "train on synthetic, test on real (TSTR)" framework [5, 9] to the sequence prediction task, we evaluate how well the generated data preserves the predictive characteristics of the original. We find that TimeGAN achieves consistent and significant improvements over state-of-the-art benchmarks in generating realistic time-series.

## 2   Related Work

TimeGAN is a generative time-series model, trained adversarially and jointly via a learned embedding space with both supervised and unsupervised losses. As such, our approach straddles the intersection of multiple strands of research, combining themes from autoregressive models for sequence prediction, GAN-based methods for sequence generation, and time-series representation learning.

Autoregressive recurrent networks trained via the maximum likelihood principle [10] are prone to potentially large prediction errors when performing multi-step sampling, due to the discrepancy between *closed-loop* training (*i.e.* conditioned on ground truths) and *open-loop* inference (*i.e.* conditioned on previous guesses). Based on curriculum learning [11], Scheduled Sampling was first proposed as a remedy, whereby models are trained to generate output conditioned on a mix of both previous guesses and ground-truth data [1]. Inspired by adversarial domain adaptation [12], Professor Forcing involved training an auxiliary discriminator to distinguish between free-running and teacher-forced hidden states, thus encouraging the network's training and sampling dynamics to converge [2]. Actor-critic methods [13] have also been proposed, introducing a critic conditioned on target outputs, trained to estimate next-token value functions that guide the actor's free-running predictions [3]. However, while the motivation for these methods is similar to ours in accounting for stepwise transition dynamics, they are inherently deterministic, and do not accommodate explicitly sampling from a learned distribution—central to our goal of synthetic data generation.

On the other hand, multiple studies have straightforwardly inherited the GAN framework within the temporal setting. The first (C-RNN-GAN) [4] directly applied the GAN architecture to sequential data, using LSTM networks for generator and discriminator. Data is generated recurrently, taking as inputs a noise vector and the data generated from the previous time step. Recurrent Conditional GAN (RCGAN) [5] took a similar approach, introducing minor architectural differences such as dropping the dependence on the previous output while conditioning on additional input [14]. A multitude of applied studies have since utilized these frameworks to generate synthetic sequences in such diverse domains as text [15], finance [16], biosignals [17], sensor [18] and smart grid data [19], as well as renewable scenarios [20]. Recent work [6] has proposed conditioning on time stamp information to

handle irregularly sampling. However, unlike our proposed technique, these approaches rely only on the binary adversarial feedback for learning, which by itself may not be sufficient to guarantee specifically that the network efficiently captures the temporal dynamics in the training data.

Finally, representation learning in the time-series setting primarily deals with the benefits of learning compact encodings for the benefit of downstream tasks such as prediction [21], forecasting [22], and classification [23]. Other works have studied the utility of learning latent representations for purposes of pre-training [24], disentanglement [25], and interpretability [26]. Meanwhile in the static setting, several works have explored the benefit of combining autoencoders with adversarial training, with objectives such as learning similarity measures [27], enabling efficient inference [28], as well as improving generative capability [29]—an approach that has subsequently been applied to generating discrete structures by encoding and generating entire sequences for discrimination [30]. By contrast, our proposed method generalizes to arbitrary time-series data, incorporates stochasticity at each time step, as well as employing an embedding network to identify a lower-dimensional space for the generative model to learn the stepwise distributions and latent dynamics of the data.

Figure 1(a) provides a high-level block diagram of TimeGAN, and Figure 2 gives an illustrative implementation, with C-RNN-GAN and RCGAN similarly detailed. For purposes of expository and experimental comparison with existing methods, we employ a standard RNN parameterization. A table of related works with additional detail can be found in the Supplementary Materials.

## 3 Problem Formulation

Consider the general data setting where each instance consists of two elements: static features (that do not change over time, e.g. gender), and temporal features (that occur over time, e.g. vital signs). Let $\mathcal{S}$ be a vector space of static features, $\mathcal{X}$ of temporal features, and let $\mathbf{S} \in \mathcal{S}, \mathbf{X} \in \mathcal{X}$ be random vectors that can be instantiated with specific values denoted $\mathbf{s}$ and $\mathbf{x}$. We consider tuples of the form $(\mathbf{S}, \mathbf{X}_{1:T})$ with some joint distribution $p$. The length $T$ of each sequence is also a random variable, the distribution of which—for notational convenience—we absorb into $p$. In the training data, let individual samples be indexed by $n \in \{1, ..., N\}$, so we can denote the training dataset $\mathcal{D} = \{(\mathbf{s}_n, \mathbf{x}_{n,1:T_n})\}_{n=1}^N$. Going forward, subscripts $n$ are omitted unless explicitly required.

Our goal is to use training data $\mathcal{D}$ to learn a density $\hat{p}(\mathbf{S}, \mathbf{X}_{1:T})$ that best approximates $p(\mathbf{S}, \mathbf{X}_{1:T})$. This is a high-level objective, and—depending on the lengths, dimensionality, and distribution of the data—may be difficult to optimize in the standard GAN framework. Therefore we additionally make use of the autoregressive decomposition of the joint $p(\mathbf{S}, \mathbf{X}_{1:T}) = p(\mathbf{S}) \prod_t p(\mathbf{X}_t | \mathbf{S}, \mathbf{X}_{1:t-1})$ to focus specifically on the conditionals, yielding the complementary—and simpler—objective of learning a density $\hat{p}(\mathbf{X}_t | \mathbf{S}, \mathbf{X}_{1:t-1})$ that best approximates $p(\mathbf{X}_t | \mathbf{S}, \mathbf{X}_{1:t-1})$ at any time $t$.

**Two Objectives**. Importantly, this breaks down the sequence-level objective (matching the joint distribution) into a series of stepwise objectives (matching the conditionals). The first is global,

$$\min_{\hat{p}} D\Big(p(\mathbf{S}, \mathbf{X}_{1:T}) \big\| \hat{p}(\mathbf{S}, \mathbf{X}_{1:T})\Big) \tag{1}$$

where $D$ is some appropriate measure of distance between distributions. The second is local,

$$\min_{\hat{p}} D\Big(p(\mathbf{X}_t | \mathbf{S}, \mathbf{X}_{1:t-1}) \big\| \hat{p}(\mathbf{X}_t | \mathbf{S}, \mathbf{X}_{1:t-1})\Big) \tag{2}$$

for any $t$. Under an ideal discriminator in the GAN framework, the former takes the form of the Jensen-Shannon divergence. Using the original data for supervision via maximum-likelihood (ML) training, the latter takes the form of the Kullback-Leibler divergence. Note that minimizing the former relies on the presence of a perfect adversary (which we may not have access to), while minimizing the latter only depends on the presence of ground-truth sequences (which we do have access to). Our target, then, will be a combination of the GAN objective (proportional to Expression 1) and the ML objective (proportional to Expression 2). As we shall see, this naturally yields a training procedure that involves the simple addition of a supervised loss to guide adversarial learning.

## 4 Proposed Model: Time-series GAN (TimeGAN)

TimeGAN consists of four network components: an embedding function, recovery function, sequence generator, and sequence discriminator. The key insight is that the autoencoding components (first two)

are trained jointly with the adversarial components (latter two), such that TimeGAN simultaneously learns to *encode* features, *generate* representations, and *iterate* across time. The embedding network provides the latent space, the adversarial network operates within this space, and the latent dynamics of both real and synthetic data are synchronized through a supervised loss. We describe each in turn.

## 4.1 Embedding and Recovery Functions

The embedding and recovery functions provide mappings between feature and latent space, allowing the adversarial network to learn the underlying temporal dynamics of the data via lower-dimensional representations. Let $\mathcal{H}_\mathcal{S}, \mathcal{H}_\mathcal{X}$ denote the latent vector spaces corresponding to feature spaces $\mathcal{S}, \mathcal{X}$. Then the embedding function $e : \mathcal{S} \times \prod_t \mathcal{X} \to \mathcal{H}_\mathcal{S} \times \prod_t \mathcal{H}_\mathcal{X}$ takes static and temporal features to their latent codes $\mathbf{h}_\mathcal{S}, \mathbf{h}_{1:T} = e(\mathbf{s}, \mathbf{x}_{1:T})$. In this paper, we implement $e$ via a recurrent network,

$$\mathbf{h}_\mathcal{S} = e_\mathcal{S}(\mathbf{s}), \qquad \mathbf{h}_t = e_\mathcal{X}(\mathbf{h}_\mathcal{S}, \mathbf{h}_{t-1}, \mathbf{x}_t) \qquad (3)$$

where $e_\mathcal{S} : \mathcal{S} \to \mathcal{H}_\mathcal{S}$ is an embedding network for static features, and $e_\mathcal{X} : \mathcal{H}_\mathcal{S} \times \mathcal{H}_\mathcal{X} \times \mathcal{X} \to \mathcal{H}_\mathcal{X}$ a recurrent embedding network for temporal features. In the opposite direction, the recovery function $r : \mathcal{H}_\mathcal{S} \times \prod_t \mathcal{H}_\mathcal{X} \to \mathcal{S} \times \prod_t \mathcal{X}$ takes static and temporal codes back to their feature representations $\tilde{\mathbf{s}}, \tilde{\mathbf{x}}_{1:T} = r(\mathbf{h}_\mathcal{S}, \mathbf{h}_{1:T})$. Here we implement $r$ through a feedforward network at each step,

$$\tilde{\mathbf{s}} = r_\mathcal{S}(\mathbf{h}_s), \qquad \tilde{\mathbf{x}}_t = r_\mathcal{X}(\mathbf{h}_t) \qquad (4)$$

where $r_\mathcal{S} : \mathcal{H}_\mathcal{S} \to \mathcal{S}$ and $r_\mathcal{X} : \mathcal{H}_\mathcal{X} \to \mathcal{X}$ are recovery networks for static and temporal embeddings. Note that the embedding and recovery functions can be parameterized by any architecture of choice, with the only stipulation being that they be autoregressive and obey causal ordering (*i.e.* output(s) at each step can only depend on preceding information). For example, it is just as possible to implement the former with temporal convolutions [31], or the latter via an attention-based decoder [32]. Here we choose implementations 3 and 4 as a minimal example to isolate the source of gains.

## 4.2 Sequence Generator and Discriminator

Instead of producing synthetic output directly in feature space, the generator first outputs into the embedding space. Let $\mathcal{Z}_\mathcal{S}, \mathcal{Z}_\mathcal{X}$ denote vector spaces over which known distributions are defined, and from which random vectors are drawn as input for generating into $\mathcal{H}_\mathcal{S}, \mathcal{H}_\mathcal{X}$. Then the generating function $g : \mathcal{Z}_\mathcal{S} \times \prod_t \mathcal{Z}_\mathcal{X} \to \mathcal{H}_\mathcal{S} \times \prod_t \mathcal{H}_\mathcal{X}$ takes a tuple of static and temporal random vectors to synthetic latent codes $\hat{\mathbf{h}}_\mathcal{S}, \hat{\mathbf{h}}_{1:T} = g(\mathbf{z}_\mathcal{S}, \mathbf{z}_{1:T})$. We implement $g$ through a recurrent network,

$$\hat{\mathbf{h}}_\mathcal{S} = g_\mathcal{S}(\mathbf{z}_\mathcal{S}), \qquad \hat{\mathbf{h}}_t = g_\mathcal{X}(\hat{\mathbf{h}}_\mathcal{S}, \hat{\mathbf{h}}_{t-1}, \mathbf{z}_t) \qquad (5)$$

where $g_\mathcal{S} : \mathcal{Z}_\mathcal{S} \to \mathcal{H}_\mathcal{S}$ is an generator network for static features, and $g_\mathcal{X} : \mathcal{H}_\mathcal{S} \times \mathcal{H}_\mathcal{X} \times \mathcal{Z}_\mathcal{X} \to \mathcal{H}_\mathcal{X}$ is a recurrent generator for temporal features. Random vector $\mathbf{z}_\mathcal{S}$ can be sampled from a distribution of choice, and $\mathbf{z}_t$ follows a stochastic process; here we use the Gaussian distribution and Wiener process

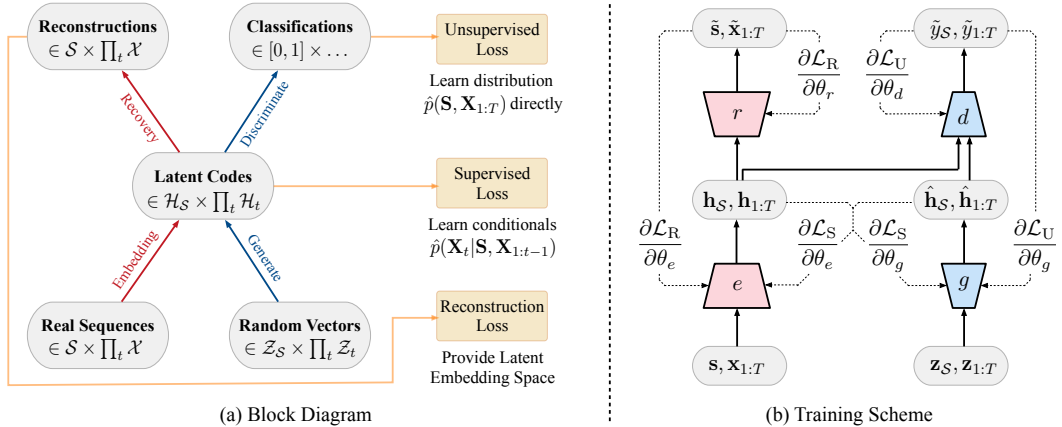

(a) Block Diagram

(b) Training Scheme

Figure 1: (a) Block diagram of component functions and objectives. (b) Training scheme; solid lines indicate forward propagation of data, and dashed lines indicate backpropagation of gradients.

respectively. Finally, the discriminator also operates from the embedding space. The discrimination function $d : \mathcal{H}_\mathcal{S} \times \prod_t \mathcal{H}_\mathcal{X} \to [0,1] \times \prod_t [0,1]$ receives the static and temporal codes, returning classifications $\tilde{y}_\mathcal{S}, \tilde{y}_{1:T} = d(\tilde{\mathbf{h}}_\mathcal{S}, \tilde{\mathbf{h}}_{1:T})$. The $\tilde{\mathbf{h}}_*$ notation denotes either real ($\mathbf{h}_*$) or synthetic ($\hat{\mathbf{h}}_*$) embeddings; similarly, the $\tilde{y}_*$ notation denotes classifications of either real ($y_*$) or synthetic ($\hat{y}_*$) data. Here we implement $d$ via a bidirectional recurrent network with a feedforward output layer,

$$\tilde{y}_\mathcal{S} = d_\mathcal{S}(\tilde{\mathbf{h}}_\mathcal{S}) \qquad \tilde{y}_t = d_\mathcal{X}(\overleftarrow{\mathbf{u}}_t, \overrightarrow{\mathbf{u}}_t) \tag{6}$$

where $\overrightarrow{\mathbf{u}}_t = \overrightarrow{c}_\mathcal{X}(\tilde{\mathbf{h}}_\mathcal{S}, \tilde{\mathbf{h}}_t, \overrightarrow{\mathbf{u}}_{t-1})$ and $\overleftarrow{\mathbf{u}}_t = \overleftarrow{c}_\mathcal{X}(\tilde{\mathbf{h}}_\mathcal{S}, \tilde{\mathbf{h}}_t, \overleftarrow{\mathbf{u}}_{t+1})$ respectively denote the sequences of forward and backward hidden states, $\overrightarrow{c}_\mathcal{X}, \overleftarrow{c}_\mathcal{X}$ are recurrent functions, and $d_\mathcal{S}, d_\mathcal{X}$ are output layer classification functions. Similarly, there are no restrictions on architecture beyond the generator being autoregressive; here we use a standard recurrent formulation for ease of exposition.

### 4.3 Jointly Learning to Encode, Generate, and Iterate

First, purely as a reversible mapping between feature and latent spaces, the embedding and recovery functions should enable accurate reconstructions $\tilde{\mathbf{s}}, \tilde{\mathbf{x}}_{1:T}$ of the original data $\mathbf{s}, \mathbf{x}_{1:T}$ from their latent representations $\mathbf{h}_\mathcal{S}, \mathbf{h}_{1:T}$. Therefore our first objective function is the *reconstruction loss*,

$$\mathcal{L}_\mathrm{R} = \mathbb{E}_{\mathbf{s}, \mathbf{x}_{1:T} \sim p} \big[ \|\mathbf{s} - \tilde{\mathbf{s}}\|_2 + \textstyle\sum_t \|\mathbf{x}_t - \tilde{\mathbf{x}}_t\|_2 \big] \tag{7}$$

In TimeGAN, the generator is exposed to two types of inputs during training. First, in pure open-loop mode, the generator—which is autoregressive—receives synthetic embeddings $\hat{\mathbf{h}}_\mathcal{S}, \hat{\mathbf{h}}_{1:t-1}$ (*i.e.* its own previous outputs) in order to generate the next synthetic vector $\hat{\mathbf{h}}_t$. Gradients are then computed on the *unsupervised loss*. This is as one would expect—that is, to allow maximizing (for the discriminator) or minimizing (for the generator) the likelihood of providing correct classifications $\hat{y}_\mathcal{S}, \hat{y}_{1:T}$ for both the training data $\mathbf{h}_\mathcal{S}, \mathbf{h}_{1:T}$ as well as for synthetic output $\hat{\mathbf{h}}_\mathcal{S}, \hat{\mathbf{h}}_{1:T}$ from the generator,

$$\mathcal{L}_\mathrm{U} = \mathbb{E}_{\mathbf{s}, \mathbf{x}_{1:T} \sim p} \big[ \log y_\mathcal{S} + \textstyle\sum_t \log y_t \big] + \mathbb{E}_{\mathbf{s}, \mathbf{x}_{1:T} \sim \hat{p}} \big[ \log(1 - \hat{y}_\mathcal{S}) + \textstyle\sum_t \log(1 - \hat{y}_t) \big] \tag{8}$$

Relying solely on the discriminator's binary adversarial feedback may not be sufficient incentive for the generator to capture the stepwise conditional distributions in the data. To achieve this more efficiently, we introduce an additional loss to further discipline learning. In an alternating fashion, we also train in closed-loop mode, where the generator receives sequences of embeddings of actual data $\mathbf{h}_{1:t-1}$ (*i.e.* computed by the embedding network) to generate the next latent vector. Gradients can now be computed on a loss that captures the discrepancy between distributions $p(\mathbf{H}_t | \mathbf{H}_\mathcal{S}, \mathbf{H}_{1:t-1})$ and $\hat{p}(\mathbf{H}_t | \mathbf{H}_\mathcal{S}, \mathbf{H}_{1:t-1})$. Applying maximum likelihood yields the familiar *supervised loss*,

$$\mathcal{L}_\mathrm{S} = \mathbb{E}_{\mathbf{s}, \mathbf{x}_{1:T} \sim p} \big[ \textstyle\sum_t \|\mathbf{h}_t - g_\mathcal{X}(\mathbf{h}_\mathcal{S}, \mathbf{h}_{t-1}, \mathbf{z}_t)\|_2 \big] \tag{9}$$

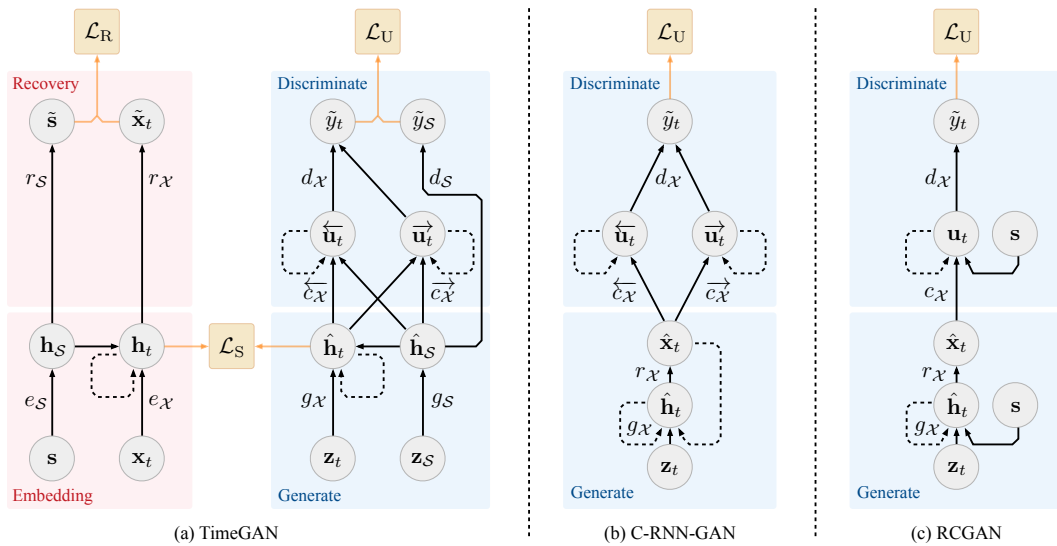

Figure 2: (a) TimeGAN instantiated with RNNs, (b) C-RNN-GAN, and (c) RCGAN. Solid lines denote function application, dashed lines denote recurrence, and orange lines indicate loss computation.

where $g_{\mathcal{X}}(\mathbf{h}_{\mathcal{S}}, \mathbf{h}_{t-1}, \mathbf{z}_t)$ approximates $\mathbb{E}_{\mathbf{z}_t \sim \mathcal{N}}[\hat{p}(\mathbf{H}_t|\mathbf{H}_{\mathcal{S}}, \mathbf{H}_{1:t-1}, \mathbf{z}_t)]$ with one sample $\mathbf{z}_t$—as is standard in stochastic gradient descent. In sum, at any step in a training sequence, we assess the difference between the actual next-step latent vector (from the embedding function) and synthetic next-step latent vector (from the generator—conditioned on the actual historical sequence of latents). While $\mathcal{L}_{\text{U}}$ pushes the generator to create realistic sequences (evaluated by an imperfect adversary), $\mathcal{L}_{\text{S}}$ further ensures that it produces similar stepwise transitions (evaluated by ground-truth targets).

**Optimization**. Figure 1(b) illustrates the mechanics of our approach at training. Let $\theta_e, \theta_r, \theta_g, \theta_d$ respectively denote the parameters of the embedding, recovery, generator, and discriminator networks. The first two components are trained on both the reconstruction and supervised losses,

$$\min_{\theta_e, \theta_r}(\lambda \mathcal{L}_{\text{S}} + \mathcal{L}_{\text{R}}) \tag{10}$$

where $\lambda \geq 0$ is a hyperparameter that balances the two losses. Importantly, $\mathcal{L}_{\text{S}}$ is included such that the embedding process not only serves to reduce the dimensions of the adversarial learning space—it is actively conditioned to facilitate the generator in learning temporal relationships from the data. Next, the generator and discriminator networks are trained adversarially as follows,

$$\min_{\theta_g}(\eta \mathcal{L}_{\text{S}} + \max_{\theta_d} \mathcal{L}_{\text{U}}) \tag{11}$$

where $\eta \geq 0$ is another hyperparameter that balances the two losses. That is, in addition to the unsupervised minimax game played over classification accuracy, the generator additionally minimizes the supervised loss. By combining the objectives in this manner, TimeGAN is simultaneously trained to encode (feature vectors), generate (latent representations), and iterate (across time).

In practice, we find that TimeGAN is not sensitive to $\lambda$ and $\eta$; for all experiments in Section 5, we set $\lambda = 1$ and $\eta = 10$. Note that while GANs in general are not known for their ease of training, we do not discover any additional complications in TimeGAN. The embedding task serves to regularize adversarial learning—which now occurs in a lower-dimensional latent space. Similarly, the supervised loss has a constraining effect on the stepwise dynamics of the generator. For both reasons, we do not expect TimeGAN to be *more* challenging to train, and standard techniques for improving GAN training are still applicable. Algorithm pseudocode and illustrations with additional detail can be found in the Supplementary Materials.

## 5  Experiments

**Benchmarks and Evaluation**. We compare TimeGAN with RCGAN [5] and C-RNN-GAN [4], the two most closely related methods. For purely autoregressive approaches, we compare against RNNs trained with teacher-forcing (T-Forcing) [33, 34] as well as professor-forcing (P-Forcing) [2]. For additional comparison, we consider the performance of WaveNet [31] as well as its GAN counterpart WaveGAN [35]. To assess the quality of generated data, we observe three desiderata: (1) *diversity*—samples should be distributed to cover the real data; (2) *fidelity*—samples should be indistinguishable from the real data; and (3) *usefulness*—samples should be just as useful as the real data when used for the same predictive purposes (i.e. train-on-synthetic, test-on-real).

**(1) Visualization**. We apply t-SNE [7] and PCA [8] analyses on both the original and synthetic datasets (flattening the temporal dimension). This visualizes how closely the distribution of generated samples resembles that of the original in 2-dimensional space, giving a qualitative assessment of (1).

**(2) Discriminative Score**. For a quantitative measure of similarity, we train a post-hoc time-series classification model (by optimizing a 2-layer LSTM) to distinguish between sequences from the original and generated datasets. First, each original sequence is labeled *real*, and each generated sequence is labeled *not real*. Then, an off-the-shelf (RNN) classifier is trained to distinguish between the two classes as a standard supervised task. We then report the classification error on the held-out test set, which gives a quantitative assessment of (2).

**(3) Predictive Score**. In order to be useful, the sampled data should inherit the predictive characteristics of the original. In particular, we expect TimeGAN to excel in capturing conditional distributions over time. Therefore, using the synthetic dataset, we train a post-hoc sequence-prediction model (by optimizing a 2-layer LSTM) to predict next-step temporal vectors over each input sequence. Then, we evaluate the trained model on the original dataset. Performance is measured in terms of the mean

absolute error (MAE); for event-based data, the MAE is computed as $|1-$ estimated probability that the event occurred|. This gives a quantitative assessment of (3).

The Supplementary Materials contains additional information on benchmarks and hyperparameters, as well as further details of visualizations and hyperparameters for the post-hoc evaluation models. Implementation of TimeGAN can be found at `https://bitbucket.org/mvdschaar/mlforhealthlabpub/src/master/alg/timegan/`.

## 5.1 Illustrative Example: Autoregressive Gaussian Models

Our primary novelties are twofold: a supervised loss to better capture temporal dynamics, and an embedding network that provides a lower-dimensional adversarial learning space. To highlight these advantages, we experiment on sequences from autoregressive multivariate Gaussian models as follows: $\mathbf{x}_t = \phi\mathbf{x}_{t-1} + \mathbf{n}$, where $\mathbf{n} \sim \mathcal{N}(\mathbf{0}, \sigma\mathbf{1} + (1-\sigma)\mathbf{I})$. The coefficient $\phi \in [0, 1]$ allows us to control the correlation across time steps, and $\sigma \in [-1, 1]$ controls the correlation across features.

As shown in Table 1, TimeGAN consistently generates higher-quality synthetic data than benchmarks, in terms of both discriminative and predictive scores. This is true across the various settings for the underlying data-generating model. Importantly, observe that the advantage of TimeGAN is greater for higher settings of temporal correlation $\phi$, lending credence to the motivation and benefit of the supervised loss mechanism. Likewise, observe that the advantage of TimeGAN is also greater for higher settings of feature correlation $\sigma$, providing confirmation for the benefit of the embedding network.

Table 1: Results on Autoregressive Multivariate Gaussian Data (Bold indicates best performance).

| Settings | Temporal Correlations (fixing $\sigma = 0.8$) | | | Feature Correlations (fixing $\phi = 0.8$) | | |
|---|---|---|---|---|---|---|
| | $\phi = 0.2$ | $\phi = 0.5$ | $\phi = 0.8$ | $\sigma = 0.2$ | $\sigma = 0.5$ | $\sigma = 0.8$ |
| Discriminative Score (Lower the better) | | | | | | |
| TimeGAN | **.175±.006** | **.174±.012** | **.105±.005** | **.181±.006** | **.152±.011** | **.105±.005** |
| RCGAN | .177±.012 | .190±.011 | .133±.019 | .186±.012 | .190±.012 | .133±.019 |
| C-RNN-GAN | .391±.006 | .227±.017 | .220±.016 | .198±.011 | .202±.010 | .220±.016 |
| T-Forcing | .500±.000 | .500±.000 | .499±.001 | .499±.001 | .499±.001 | .499±.001 |
| P-Forcing | .498±.002 | .472±.008 | .396±.018 | .460±.003 | .408±.016 | .396±.018 |
| WaveNet | .337±.005 | .235±.009 | .229±.013 | .217±.010 | .226±.011 | .229±.013 |
| WaveGAN | .336±.011 | .213±.013 | .230±.023 | .192±.012 | .205±.015 | .230±.023 |
| Predictive Score (Lower the better) | | | | | | |
| TimeGAN | **.640±.003** | **.412±.002** | **.251±.002** | **.282±.005** | **.261±0.002** | **.251±.002** |
| RCGAN | .652±.003 | .435±.002 | .263±.003 | .292±.003 | .279±.002 | .263±.003 |
| C-RNN-GAN | .696±.002 | .490±.005 | .299±.002 | .293±.005 | .280±.006 | .299±.002 |
| T-Forcing | .737±.022 | .732±.012 | .503±.037 | .515±.034 | .543±.023 | .503±.037 |
| P-Forcing | .665±.004 | .571±.005 | .289±.003 | .406±.005 | .317±.001 | .289±.003 |
| WaveNet | .718±.002 | .508±.003 | .321±.005 | .331±.004 | .297±.003 | .321±.005 |
| WaveGAN | .712±.003 | .489±.001 | .290±.002 | .325±.003 | .353±.001 | .290±.002 |

## 5.2 Experiments on Different Types of Time Series Data

We test the performance of TimeGAN across time-series data with a variety of different characteristics, including periodicity, discreteness, level of noise, regularity of time steps, and correlation across time and features. The following datasets are selected on the basis of different combinations of these properties (detailed statistics of each dataset can be found in the Supplementary Materials).

**(1) Sines**. We simulate multivariate sinusoidal sequences of different frequencies $\eta$ and phases $\theta$, providing continuous-valued, periodic, multivariate data where each feature is independent of others. For each dimension $i \in \{1, ..., 5\}$, $x_i(t) = \sin(2\pi\eta t + \theta)$, where $\eta \sim \mathcal{U}[0, 1]$ and $\theta \sim \mathcal{U}[-\pi, \pi]$.

**(2) Stocks**. By contrast, sequences of stock prices are continuous-valued but aperiodic; furthermore, features are correlated with each other. We use the daily historical Google stocks data from 2004 to 2019, including as features the volume and high, low, opening, closing, and adjusted closing prices.

**(3) Energy**. Next, we consider a dataset characterized by noisy periodicity, higher dimensionality, and correlated features. The UCI Appliances energy prediction dataset consists of multivariate, continuous-valued measurements including numerous temporal features measured at close intervals.

**(4) Events**. Finally, we consider a dataset characterized by discrete values and irregular time stamps. We use a large private lung cancer pathways dataset consisting of sequences of events and their times, and model both the one-hot encoded sequence of event types as well as the event timings.

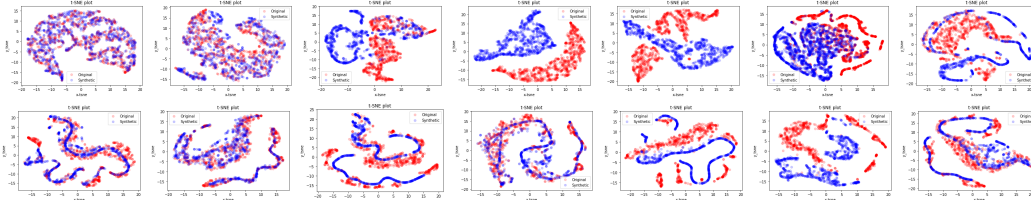

(a) TimeGAN   (b) RCGAN   (c) CRNNGAN   (d) T-Forcing   (e) P-Forcing   (f) WaveNet   (g) WaveGAN

Figure 3: t-SNE visualization on Sines (1st row) and Stocks (2nd row). Each column provides the visualization for each of the 7 benchmarks. Red denotes original data, and blue denotes synthetic. Additional and larger t-SNE and PCA visualizations can be found in the Supplementary Materials.

**Visualizations with t-SNE and PCA**. In Figure 3, we observe that synthetic datasets generated by TimeGAN show markedly better overlap with the original data than other benchmarks using t-SNE for visualization (PCA analysis can be found in the Supplementary Materials). In fact, we (in the first column) that the blue (generated) samples and red (original) samples are almost perfectly in sync.

**Discriminative and Predictive Scores**. As indicated in Table 2, TimeGAN consistently generates higher-quality synthetic data in comparison to benchmarks on the basis of both discriminative (post-hoc classification error) and predictive (mean absolute error) scores across all datasets. For instance for Stocks, TimeGAN-generated samples achieve 0.102 which is 48% lower than the next-best benchmark (RCGAN, at 0.196)—a statistically significant improvement. Remarkably, observe that the predictive scores of TimeGAN are almost on par with those of the original datasets themselves.

Table 2: Results on Multiple Time-Series Datasets (Bold indicates best performance).

| Metric | Method | Sines | Stocks | Energy | Events |
|---|---|---|---|---|---|
| Discriminative Score (Lower the Better) | TimeGAN | **.011±.008** | **.102±.021** | **.236±.012** | **.161±.018** |
| | RCGAN | .022±.008 | .196±.027 | .336±.017 | .380±.021 |
| | C-RNN-GAN | .229±.040 | .399±.028 | .499±.001 | .462±.011 |
| | T-Forcing | .495±.001 | .226±.035 | .483±.004 | .387±.012 |
| | P-Forcing | .430±.027 | .257±.026 | .412±.006 | .489±.001 |
| | WaveNet | .158±.011 | .232±.028 | .397±.010 | .385±.025 |
| | WaveGAN | .277±.013 | .217±.022 | .363±.012 | .357±.017 |
| Predictive Score (Lower the Better) | TimeGAN | **.093±.019** | **.038±.001** | **.273±.004** | **.303±.006** |
| | RCGAN | .097±.001 | .040±.001 | .292±.005 | .345±.010 |
| | C-RNN-GAN | .127±.004 | **.038±.000** | .483±.005 | .360±.010 |
| | T-Forcing | .150±.022 | **.038±.001** | .315±.005 | .310±.003 |
| | P-Forcing | .116±.004 | .043±.001 | .303±.006 | .320±.008 |
| | WaveNet | .117±.008 | .042±.001 | .311±.005 | .333±.004 |
| | WaveGAN | .134±.013 | .041±.001 | .307±.007 | .324±.006 |
| | Original | .094±.001 | .036±.001 | .250±.003 | .293±.000 |

### 5.3 Sources of Gain

TimeGAN is characterized by (1) the supervised loss, (2) embedding networks, and (3) the joint training scheme. To analyze the importance of each contribution, we report the discriminative and predictive scores with the following modifications to TimeGAN: (1) without the supervised loss, (2) without the embedding networks, and (3) without jointly training the embedding and adversarial networks on the supervised loss. (The first corresponds to $\lambda = \eta = 0$, and the third to $\lambda = 0$).

Table 3: Source-of-Gain Analysis on Multiple Datasets (via Discriminative and Predictive scores).

| Metric | Method | Sines | Stocks | Energy | Events |
|---|---|---|---|---|---|
| Discriminative Score (Lower the Better) | TimeGAN | **.011**±**.008** | **.102**±**.021** | **.236**±**.012** | **.161**±**.018** |
| | w/o Supervised Loss | .193±.013 | .145±.023 | .298±.010 | .195±.013 |
| | w/o Embedding Net. | .197±.025 | .260±.021 | .286±.006 | .244±.011 |
| | w/o Joint Training | .048±.011 | .131±.019 | .268±.012 | .181±.011 |
| Predictive Score (Lower the Better) | TimeGAN | **.093**±**.019** | **.038**±**.001** | **.273**±**.004** | **.303**±**.006** |
| | w/o Supervised Loss | .116±.010 | .054±.001 | .277±.005 | .380±.023 |
| | w/o Embedding Net. | .124±.002 | .048±.001 | .286±.002 | .410±.013 |
| | w/o Joint Training | .107±.008 | .045±.001 | .276±.004 | .348±.021 |

We observe in Table 3 that all three elements make important contributions in improving the quality of the generated time-series data. The supervised loss plays a particularly important role when the data is characterized by high temporal correlations, such as in the Stocks dataset. In addition, we find that the embedding networks and joint training the with the adversarial networks (thereby aligning the targets of the two) clearly and consistently improves generative performance across the board.

## 6 Conclusion

In this paper we introduce TimeGAN, a novel framework for time-series generation that combines the versatility of the unsupervised GAN approach with the control over conditional temporal dynamics afforded by supervised autoregressive models. Leveraging the contributions of the supervised loss and jointly trained embedding network, TimeGAN demonstrates consistent and significant improvements over state-of-the-art benchmarks in generating realistic time-series data. In the future, further work may investigate incorporating the differential privacy framework into the TimeGAN approach in order to generate high-quality time-series data with differential privacy guarantees.

## Acknowledgements

The authors would like to thank the reviewers for their helpful comments. This work was supported by the National Science Foundation (NSF grants 1407712, 1462245 and 1533983), and the US Office of Naval Research (ONR).

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
