[Supplementary Material]

# Supplementary Materials:
# Time-series Generative Adversarial Networks

**Jinsung Yoon**[*]
University of California, Los Angeles, USA
jsyoon0823@g.ucla.edu

**Daniel Jarrett**[*]
University of Cambridge, UK
daniel.jarrett@maths.cam.ac.uk

**Mihaela van der Schaar**
University of Cambridge, UK
University of California, Los Angeles, USA
Alan Turing Institute, UK
mv472@cam.ac.uk, mihaela@ee.ucla.edu

## Additional Related Work

TimeGAN integrates ideas from autoregressive models for sequence prediction [1, 2, 3], GAN-based methods for sequence generation [4, 5, 6], and time-series representation learning [7, 8, 9]—the relation and details for which are discussed in the main manuscript. In this section, we additionally discuss methods related on the periphery, including RNN-based sequence models using variational autoencoders, as well as GAN-based approaches for semi-supervised learning.

RNN-based models have been combined with variational autoencoders to generate sequences. In [10], this was done by learning to map *entire sequences* to single latent vectors, with the goal of capturing high-level properties of sequences and interpolating in latent space. This idea was extended to the general time-series setting [11], with the additional proposal that the trained weights and network states can be used to initialize standard RNN models. However, in both cases sampling from the prior over these representations involved but a simple deterministic decoder, so the only source of variability is found in the conditional output probability model. On the other hand, [12] proposed augmenting the representational power of the standard RNN model with stochastic latent variables *at each time step*. Recognizing that model variability should induce dependencies across time steps, [13] further extended this approach to accommodate temporal dependencies between latent random variables, and [14] explicitly layer a state space model on top of the RNN structure. In parallel, this technique has since been applied to temporal convolutional models of sequences as well, with stochastic latent variables injected into the WaveNet structure [15]. However, the focus of these methods is specifically on encoding sufficient input variability to model highly structured data (such as speech and handwriting). In particular, they do not ensure that new sequences unconditionally sampled from the model match the underlying distribution of the training data. By contrast, TimeGAN focuses on learning the entire (joint) distribution such that sampled data match the original, while simultaneously ensuring that the model respects the (conditional) dynamics of the original data.

It is also worth mentioning that our method bears superficial resemblance to GAN-based approaches for semi-supervised learning. With methods such as [18, 19, 20], the task of interest is one of supervised classification, with an auxiliary unsupervised loss for generating additional unlabeled examples for training. Conversely, the focus of TimeGAN is on the unsupervised task of generative modeling, with an auxiliary supervised loss to provide additional control over the network's dynamics.

---

[*] indicates equal contribution

Table 1: Summary of Related Work. (Open-loop: Previous outputs are used as conditioning information for generation at each step; Mixed-variables: Accommodates static & temporal variables).

| | C-RNN-GAN [4] | RCGAN [5] | T-Forcing [16, 17] | P-Forcing [2] | TimeGAN (Ours) |
|---|:---:|:---:|:---:|:---:|:---:|
| Stochastic | ✓ | ✓ | | | ✓ |
| Open-loop | ✓ | | ✓ | ✓ | ✓ |
| Adversarial loss | ✓ | ✓ | | ✓ | ✓ |
| Supervised loss | | | ✓ | ✓ | ✓ |
| Discrete features | | | | ✓ | ✓ |
| Embedding space | | | | | ✓ |
| Mixed-variables | | | | | ✓ |

## Additional Illustrations

Figure 1(b) in the main manuscript details the training scheme for TimeGAN. For side-by-side comparison, Figure 1(a) below additionally illustrates the training scheme for existing methods C-RNN-GAN and RCGAN, which employ the standard GAN setup during training. Furthermore, Figure 1(b) also compares the flow of data during sampling time for TimeGAN and these existing methods.

(a) Training Time

(b) Sampling Time

Figure 1: Block diagram of our proposed method (TimeGAN), shown here in comparison with existing methods (C-RNN-GAN and RCGAN) during (a) training time, as well as (b) sampling time. Solid lines indicate forward propagation of data, and dashed lines indicate backpropagation of gradients.

## Hyperparameters and Benchmarks

We use tensorflow to implement TimeGAN; source code will be made available after acceptance. All of the components (embedding network, generator, and discriminator) are implemented with 3-layer GRUs with hidden dimensions 4 times the size of the input features. The dimension of the latent space is half that of the input features. We use tanh as the activation function and sigmoid as the output layer activation function such that outputs are within the $[0, 1]$ range. We also normalize the dataset to the $[0, 1]$ range using min-max scaling. We set $\lambda = 1$ and $\eta = 10$ in our experiments.

We use the following publicly available source code to implement our benchmarks.

- C-RNN-GAN [4]: https://github.com/olofmogren/c-rnn-gan
- RCGAN [5]: https://github.com/ratschlab/RGAN

- T-Forcing [16]: `https://github.com/snowkylin/rnn-handwriting-generation`
- P-Forcing [2]: `https://github.com/anirudh9119/LM_GANS`
- WaveNet [21]: `https://github.com/ibab/tensorflow-wavenet`
- WaveGAN [22]: `https://github.com/chrisdonahue/wavegan`

For fair comparison, we use the same underlying recurrent neural network architecture (3-layer GRUs with hidden dimensions 4 times the size of input features) for C-RNN-GAN, RCGAN, T-Forcing, and P-Forcing as is used in TimeGAN. In the case of deterministic models (such as T-Forcing and P-Forcing), we first train an original GAN model to generate feature vectors as inputs for the initial time step, which follows the original feature distribution at the initial time step. Then, using the generated feature vector as input, we initialize the model to generate the sequence in open-loop mode. Finally, the post-hoc time-series classification and sequence-prediction models are implemented as 2-layer LSTMs with hidden dimensions 4 times the size of the input features. As before, we use tanh as the activation function and sigmoid as the output layer activation function such that outputs are within the $[0, 1]$ range.

## Additional Dataset Statistics

Table 2: Additional Dataset Statistics

| Dataset | Sequences | Dim. | Avg. Len. | Feature Corr. | Temporal Variance | Temporal Corr. |
|---|---|---|---|---|---|---|
| Sines | 10,000 | 5 | 24 pts | 0.0117 | 0.3167 | 0.2056 |
| Stocks | 3,773 | 6 | 24 days | 0.8596 | 0.0129 | 0.9902 |
| Energy | 19,711 | 29 | 24 hrs | 0.2843 | 0.0444 | 0.8506 |
| Events | 149,967 | 54 | 58 events | 0.0095 | 0.0622 | 0.0744 |

The Google Stocks dataset is available online, and can be downloaded from: `LINK`. The UCI Appliances Energy Prediction dataset is also available online, and can be downloaded from: `LINK`.

## Algorithm Pseudocode

---

**Algorithm 1** Pseudocode of TimeGAN

---

1: **Input:** $\lambda = 1$, $\eta = 10$, $\mathcal{D}$, batch size $n_{mb}$, learning rate $\gamma$
2: **Initialize:** $\theta_e, \theta_r, \theta_g, \theta_d$
3: **while** Not converged **do**
4:    **(1) Map between Feature Space and Latent Space**
5:        Sample $(\mathbf{s}_1, \mathbf{x}_{1,1:T_n}), ..., (\mathbf{s}_{n_{mb}}, \mathbf{x}_{n_{mb},1:T_{n_{mb}}}) \overset{\text{i.i.d.}}{\sim} \mathcal{D}$
6:        **for** $n = 1, ..., n_{mb}, t = 1, ..., T_n$ **do**
7:            $(\mathbf{h}_{n,\mathcal{S}}, \mathbf{h}_{n,t}) = (e_{\mathcal{S}}(\mathbf{s}_n), e_{\mathcal{X}}(\mathbf{h}_{n,\mathcal{S}}, \mathbf{h}_{n,t-1}, \mathbf{x}_{n,t}))$
8:            $(\tilde{\mathbf{s}}_n, \tilde{\mathbf{x}}_{n,t}) = (r_{\mathcal{S}}(\mathbf{h}_{n,\mathcal{S}}), r_{\mathcal{X}}(\mathbf{h}_{n,t}))$
9:
10:    **(2) Generate Synthetic Latent Codes**
11:        Sample $(\mathbf{z}_{\mathcal{S},1}, \mathbf{z}_{1,1:T_n}), ..., (\mathbf{z}_{\mathcal{S},n_{mb}}, \mathbf{z}_{n_{mb},1:T_{n_{mb}}}) \overset{\text{i.i.d.}}{\sim} p_{\mathcal{Z}_{\mathcal{S} \times \mathcal{X}}}$
12:        **for** $n = 1, ..., n_{mb}, t = 1, ..., T_n$ **do**
13:            $(\hat{\mathbf{h}}_{n,\mathcal{S}}, \hat{\mathbf{h}}_{n,t}) = (g_{\mathcal{S}}(\mathbf{z}_{\mathcal{S},n}), g_{\mathcal{X}}(\hat{\mathbf{h}}_{n,\mathcal{S}}, \hat{\mathbf{h}}_{n,t-1}, \mathbf{z}_{n,t}))$
14:
15:    **(3) Distinguish between Real and Synthetic Codes**
16:        **for** $n = 1, ..., n_{mb}, t = 1, ..., T_n$ **do**
17:            $(y_{n,\mathcal{S}}, y_{n,t}) = (d_{\mathcal{S}}(\mathbf{h}_{n,\mathcal{S}}), d_{\mathcal{X}}(\bar{\mathbf{u}}_{n,t}, \vec{\mathbf{u}}_{n,t}))$
18:            $(\hat{y}_{n,\mathcal{S}}, \hat{y}_{n,t}) = (d_{\mathcal{S}}(\hat{\mathbf{h}}_{n,\mathcal{S}}), d_{\mathcal{X}}(\bar{\hat{\mathbf{u}}}_{n,t}, \vec{\hat{\mathbf{u}}}_{n,t}))$
19:
20:    **(4) Compute Reconstruction ($\hat{\mathcal{L}}_R$), Unsupervised ($\hat{\mathcal{L}}_U$), and Supervised ($\hat{\mathcal{L}}_S$) Losses**
21:        $\hat{\mathcal{L}}_R = \frac{1}{n_{mb}} \sum_{n=1}^{n_{mb}} \left[ \|\mathbf{s}_n - \tilde{\mathbf{s}}_n\|_2 + \sum_t \|\mathbf{x}_{n,t} - \tilde{\mathbf{x}}_{n,t}\|_2 \right]$
22:        $\hat{\mathcal{L}}_U = \frac{1}{n_{mb}} \sum_{n=1}^{n_{mb}} \left[ \left[ \log y_{n,\mathcal{S}} + \sum_t \log y_{n,t} \right] + \left[ \log(1 - \hat{y}_{n,\mathcal{S}}) + \sum_t \log(1 - \hat{y}_{n,t}) \right] \right]$
23:        $\hat{\mathcal{L}}_S = \frac{1}{n_{mb}} \sum_{n=1}^{n_{mb}} \left[ \sum_t \|\mathbf{h}_{n,t} - g_{\mathcal{X}}(\mathbf{h}_{n,\mathcal{S}}, \mathbf{h}_{n,t-1}, \mathbf{z}_{n,t})\|_2 \right]$
24:
25:    **(5) Update $\theta_e, \theta_r, \theta_g, \theta_d$ via Stochastic Gradient Descent (SGD)**
26:        $\theta_e = \theta_e - \gamma \nabla_{\theta_e} - \left[ \lambda \hat{\mathcal{L}}_S + \hat{\mathcal{L}}_R \right]$
27:        $\theta_r = \theta_r - \gamma \nabla_{\theta_r} - \left[ \lambda \hat{\mathcal{L}}_S + \hat{\mathcal{L}}_R \right]$
28:        $\theta_g = \theta_g - \gamma \nabla_{\theta_g} - \left[ \eta \hat{\mathcal{L}}_S + \hat{\mathcal{L}}_U \right]$
29:        $\theta_d = \theta_d + \gamma \nabla_{\theta_d} - \hat{\mathcal{L}}_U$
30:
31: **(6) Synthetic Data Generation**
32: *(6-1) Sample* $(\mathbf{z}_{\mathcal{S},1}, \mathbf{z}_{1,1:T_n}), ..., (\mathbf{z}_{\mathcal{S},N}, \mathbf{z}_{N,1:T_N}) \overset{\text{i.i.d.}}{\sim} p_{\mathcal{Z}_{\mathcal{S} \times \mathcal{X}}}$
33: *(6-2) Generate synthetic latent codes*
34: **for** $n = 1, ..., N, t = 1, ..., T_n$ **do**
35:    $(\hat{\mathbf{h}}_{n,\mathcal{S}}, \hat{\mathbf{h}}_{n,t}) = (g_{\mathcal{S}}(\mathbf{z}_{\mathcal{S},n}), g_{\mathcal{X}}(\hat{\mathbf{h}}_{n,\mathcal{S}}, \hat{\mathbf{h}}_{n,t-1}, \mathbf{z}_{n,t}))$
36: *(6-3) Mapping to the feature space*
37: **for** $n = 1, ..., N, t = 1, ..., T_n$ **do**
38:    $(\hat{\mathbf{s}}_n, \hat{\mathbf{x}}_{1:T_n}) = (r_{\mathcal{S}}(\mathbf{h}_{n,\mathcal{S}}), r_{\mathcal{X}}(\mathbf{h}_{n,t}))$
39:
40: **Output:** $\hat{\mathcal{D}} = \{\hat{\mathbf{s}}_n, \hat{\mathbf{x}}_{1:T_n}\}_{n=1}^N$

---

# Additional Visualizations with t-SNE and PCA

Figure 2: t-SNE (1st column) and PCA (2nd column) visualizations on Sines, and t-SNE (3rd column) and PCA (4th column) visualizations on Stocks. Each row provides the visualization for each of the 7 benchmarks, ordered as follows: (1) TimeGAN, (2) RCGAN, (3) C-RNN-GAN, (4) T-Forcing, (5) P-Forcing, (6) WaveNet, and (7) WaveGAN. Red denotes original data, and blue denotes synthetic.