[Reviews · NeurIPS 2019]

Reviewer 1



Originality: The work appears original to me, but it is not the first on GANs using temporal data. There is previous work on temporal GANs used to generate video sequences (Masaki Saito et al, Temporal Generative Adversarial Nets with Singular Value Clipping). This previous work also calls their approach TGAN, but appears to be different. It may be better to use a different name for the method presented in this work. It would be good to point out differences between the approaches. Quality: The submission appears technically sound to me, I didn't check every detail though. The evaluation uses a number of different approaches for comparison, all of which perform worse than the newly introduced method. From the description it sounds the approach is mostly used/useful for datasets with only small number of variates, but hasn't been used on eg video data. In terms of evaluating weaknesses of the approach, it may be interesting to do that. Clarity: The paper is well written but clarity could be improved in several cases: - I found the notation / the explicit split between "static" and temporal features into two variables confusing, at least initially. In my view this requires more information than is provided in the paper (what is S and Xt). - even with the pseudocode given in the supplementary material I don't get the feeling the paper is written to be reproduced. It is written to provide an intuitive understanding of the work, but to actually reproduce it, more details are required that are neither provided in the paper nor in the supplementary material. This includes, for example, details about the RNN implementation (like number of units etc), and many other technical details. - the paper is presented well, e.g., quality of graphs is good (though labels on the graphs in Fig 3 could be slightly bigger) Significance: - from just the paper: the results would be more interesting (and significant) if there was a way to reproduce the work more easily. At present I cannot see this work easily taken up by many other researchers mainly due to lack of detail in the description. The work is interesting, and I like the idea, but with a relatively high-level description of it in the paper it would need a little more than the peudocode in the materials to convince me using it (but see next). - In the supplementary material it is stated the source code will be made available, and in combination with paper and information in the supplementary material, the level of detail may be just right (but it's hard to say without seeing the code). Given the promising results, I can imagine this approach being useful at least for more research in a similar direction.

Reviewer 2



1. This may be a good idea, but the evaluation is too simple to meet the acceptance threshold. I think this may be a good technique, but without extensive evaluation, it is not convincing. All the experiments are performed on UCI/Synthetic datasets, instead of larger benchmarks like previous work, e.g., RCGAN. 2. In your main text, I do not see you explain what S and X is, and I need to guess. There are many unclear points in your paper, that I need to refer related works to figure it out. 3. The evaluation metrics are not convincing, with two trained models. This is a critical issue, please find a way to fix it.

Reviewer 3



In this paper, the authors present a new generative model for time series data. The approach is based on GANs, with three key parts: 1) a supervised loss, 2) a reconstruction loss and 3) a joint training between the embedding and adversarial networks. To my knowledge, the TGAN approach is novel. It has the potential to be widely applicable to many time series problems. The paper is extremely well written and a pleasure to read. I commend the authors for explaining the technical details in a very clear manner. Figures 1 and 2 are particularly helpful at illustrating the key concepts of the paper. The evaluation is very well done. A standard evaluation section for a GAN paper often only shows examples of data generated by the GAN model (e.g. images). This typical approach is very qualitative and it is refreshing to see the authors include quantitative results from different types of experiments. There could be minor quibbles with each type of experimental setup, but as a whole, the empirical evidence is compelling. My main concern with the approach is that training GANs can be challenging. Does the training process for TGAN involve similar difficulties like mode collapse and having to give the discriminator more optimization steps than the generator during training? In addition, how sensitive is the performance of TGAN to parameters lambda and nu? Adding more parameters to an already notoriously difficult training optimization makes me nervous. The paper could be strengthened with a brief discussion of these issues. Comments after author feedback ------------------------------------------ The authors have done a good job of addressing my concerns. My review remains the same and I still feel the paper should be accepted.

[Author Response · NeurIPS 2019]

We thank all reviewers for insightful comments. *All added experiments are tabulated in revised manuscript/appendices*.

**[Reviewer 1]** • **Saito et al.**: We will rename our framework to TimeGAN to minimize confusion with [Saito, ICCV
2017], which operate *within* the standard GAN framework, proposing a special 2-stage generator (detail in revised
appendices). By contrast, we propose a different GAN *framework* altogether, where adversarial learning occurs in the
(jointly-optimized) latent space itself. • **Number of variates in experiment data**: While TimeGAN components can
indeed be instantiated with various architectures, we focus on the time series setting, using RNNs to illustrate consistent
improvement across a variety of data. Media-specific domain applications (e.g. video) are beyond the paper's scope.
However, we agree an even higher-dimensional validation is beneficial. We have conducted additional experiments
on UCI Human Activity Recognition ($dim = 561$). In short, TimeGAN achieves 0.062 (21.2%) & 0.012 (12.7%)
discriminative & predictive gains relative to the best benchmark (RCGAN). • **Static vs. temporal features**: We will
clarify the following before lines 106-107: "Consider the general data setting where each instance consists of two
elements: static features (that do not change over time, e.g. ethnicity, gender), and temporal features (that occur over
time, e.g. vital signs, clinical events)." Accommodating static features gives the most general framework, since they
often accompany temporal data (e.g. patient data). However, static features are *not* required (we can simply drop the
non-recurrent parts of $e, r, g, d$); the novelties of TimeGAN are in how it handles temporal aspects. • **Further details**
**on architecture**: We will publish full source code for TimeGAN with the camera-ready manuscript; this will contain
all specifications, training settings, and parameters necessary for reproducibility. Furthermore, in addition to lines 37-59
in the appendices, we will tabulate all technical information with the same granular level of architectural detail (down
to dimensions of individual variables) as Appendix B in [Lucic, ICML 2019]. Moreover, for additional sensitivities on
hyperparameters $\lambda, \eta$, see response Hyperparameter sensitivities for Reviewer 3. • **Discriminative metric**: To quantify
the fidelity of synthetic samples (among other desiderata; see response Evaluation metrics... for Reviewer 2), we use the
discriminative metric to gauge how indistinguishable samples are from actual data. First, actual sequences are labeled
"real", and sampled sequences "not real". Then, an off-the-shelf (RNN) classifier is trained to distinguish between the
two (a standard supervised task). We are not doing any *pairwise* testing for differences between individual sequences.

**[Reviewer 2]** • **Evaluation metrics, datasets, and benchmarks**: In the familiar application of GANs to images, the
vast majority of evaluation relies on inception scores and variants, as well as visual fidelity by inspection; importantly,
observe that the former is based on a separately trained model [Salimans, NIPS 2016]. This approach is virtually
universal [Lucic, ICML 2019; Brock, ICLR 2019; Wang, ECCV 2018]; furthermore, using post-hoc classifiers for
the evaluation of generative models is well-established [Isola, CVPR 2017; Zhang, ECCV 2016]. In the context of
time-series GANs, we observe *three* comprehensive desiderata: (1) *fidelity*—samples should be indistinguishable from
real data; (2) *diversity*—samples should be distributed to cover the real; and (3) *usefulness*—samples should be just as
useful as real data when used for the same predictive purposes (i.e. train-on-synthetic, test-on-real). In our evaluation,
the discriminative score, t-SNE/PCA analyses, as well as predictive score respectively give measures of (1), (2), and (3).

Our approach to evaluation is *much more comprehensive* than prior works on GANs for time series. First, they do not
address (1) and (2) directly. C-RNN-GAN uses a single dataset, relying on hand-crafted measures of audio fidelity.
RCGAN uses a post-hoc classifier to evaluate usefulness of samples (i.e. (3)), tested on MNIST and a single real
dataset (very low $dim = 4$); other metrics are only applied to synthetic data. By contrast, we focus on all 3 desiderata.
Second, we provide experimental results across 6 competing benchmarks over all metrics; RCGAN and C-RNN-GAN
provide zero. Third, our 5 datasets are specifically picked to vary with respect to dimensions, correlations, periodicity,
discreteness, etc. (see lines 239-254), including a massive ($n = 150k, dim = 54$) real-world medical dataset (Events).
For these reasons, we submit that our approach to evaluation is *more* comprehensive—especially w.r.t. RCGAN as the
reviewer mentions. Furthermore, see response Number of variates in experiment data for Reviewer 1 for additional
results on even higher-dimensional ($dim = 561$) data. • **Static vs. temporal features**: Kindly refer to response Static
vs. temporal features for Reviewer 1. • **Discrete data**: We already use discrete data. Our largest real-world dataset
($dim = 54$) is discrete, with $\sim 150k$ sequences (see lines 252-254, and Table 2 in appendices). TimeGAN significantly
outperforms all benchmarks on both discriminative/predictive scores (as with all datasets. See Table 2 in manuscript).

**[Reviewer 3]** • **Difficulty of training**: Although GANs in general are not the easiest to train, we did not discover any
additional complications in our experiments. The embedding task serves to regularize adversarial learning—which
now occurs in a lower-dimensional latent space; similarly, the supervised loss has a constraining effect on the stepwise
dynamics of the generator. For both reasons, we do not expect TimeGAN to be *more* challenging to train; standard
techniques for improving GAN training still apply. Here, we use covariance feature matching across all models to
improve the diversity of generation. Kindly refer to response Number of variates in experiment data for Reviewer 1
for equally favorable results on even higher-dimensional data. See also the following response. • **Hyperparameter**
**sensitivities**: We find empirically that TGAN is not very sensitive to $\lambda$ and $\eta$. While we set $\lambda = 1, \eta = 10$ for all
experiments, we agree that showing additional sensitivities may be beneficial. We have conducted experiments across a
range of hyperparameters, tabulated in the revised appendices. For example (for Stocks), across $\lambda = \{1, 5, 10, 20\}$ and
$\eta = \{0.1, 0.5, 1, 2, 5\}$, the min and max discriminative scores are 0.097 and 0.108, with variance 0.004—showing that
performance is by no means brittle—thereby providing further reassurance that TimeGAN is not more difficult to train.

[Meta-Review · NeurIPS 2019]

The reviewers agree that this work is novel and interesting, and makes an interesting contribution to the literature. Please take the reviewer comments into account while preparing the camera-ready version.